# Perinatal healthcare for women at risk of children's social care involvement: a qualitative survey of professionals in England

Claire Grant [1], Tamsin Bicknell-Morel,[2] Billie Lever Taylor [3], Claire Powell [4], Ruth Marion Blackburn [4], Rebecca Lacey,[1,5] Jenny Woodman [6]

For numbered affiliations see end of article.

**Correspondence to**
Claire Grant;
claire.grant.20@ucl.ac.uk

## ABSTRACT

**Background** Women with complex health needs are more at risk of having children's social care involvement with their newborns than other mothers. Around the time of pregnancy, there are opportunities for health services to support women with these needs and mitigate the risk of mother–baby separation. Yet little is known about healthcare professionals' experiences of providing this support.

**Methods** We administered an online survey to perinatal healthcare professionals across England (n=70 responders), including midwives, obstetricians, perinatal psychologists/psychiatrists and health visitors. We asked about their experiences of providing care for pregnant women with chronic physical conditions, mental health needs, intellectual/developmental disabilities and substance use disorders, who might be at risk of children's social care involvement. We conducted a framework analysis.

**Results** We constructed five themes from participant data. These include (1) inaccessible healthcare for women with complex needs, (2) the challenges and importance of restoring trust, (3) services focusing on individuals, not families, (4) the necessity and caution around multidisciplinary support and (5) underfunded services inhibiting good practice.

**Conclusions** Women who are at risk of children's social care involvement will likely experience perinatal healthcare inequities. Our findings suggest that current perinatal healthcare provision for this population is inadequate and national guidelines need updated to inform support.

## STRENGTHS AND LIMITATIONS OF THIS STUDY

⇒ This study offers a timely insight into lived experiences of providing perinatal healthcare for women at risk of children's social care involvement.
⇒ We collected survey data across different local authorities in England and across a range of professionals who support pregnant women in healthcare.
⇒ Free-text responses allowed professionals to share in-depth responses and case examples of service provision and service constraints.
⇒ Survey methods can be bias towards those who 'self-select' to participate and have a given motive for doing so (eg, a strong view that they wish to share).
⇒ Findings can contribute towards the development of national guidelines for supporting women at risk of children's social care involvement.

## BACKGROUND

From conception until 1 year after birth (the 'perinatal' period), mothers in England have various routine contacts with healthcare services. As well as labour and birth support, most women are offered up to 10 antenatal appointments to check the health and development of their babies, in addition to regular postnatal reviews from health visitors, and specialist input, such as from perinatal mental health services, as needed.[1]

Mothers with complex health needs are more likely to have complications during pregnancy and childbirth.[2–4] Newborns are directly affected by parental health and health behaviours, via in utero effects to the developing baby and postnatal caregiving practices.[5] There are also indirect harms of poor parental health on infants, including unstable housing, compromised parental capacity and increased risk of child abuse/neglect.[6] It is known that histories of trauma, experiences of chronic stress and psychosocial adversities can impact on women's experiences of birth and infant outcomes.[7 8] Women who experience severe and multiple disadvantages can be underserved by maternity services and overrepresented in children's social care.[9 10]

In a 2022 review of perinatal mortality in the UK, 20% of women who died during pregnancy, or in the year following, were found to be involved with children's social care.[11] Maternal serious mental illness,[12] substance use disorder[13] and intellectual/developmental disability[14] have all been

associated with an increased risk of prebirth assessments and newborn entry into out-of-home care.[10] The fear of children's social care involvement can itself act as a barrier for parents accessing help from relevant services, including specialist support for their mental health or substance use.[15 16] Additionally, health needs can coexist and be compounded by adverse social circumstances, such as low social support and living in poverty.[17 18]

Many of the challenges faced by women with children's social care involvement are the product of lifelong disadvantages and lack of meaningful support,[19] factors which long predate their pregnancy.[7] There are also various barriers to providing adequate care for this population, including parental distrust of services.[15] Nevertheless, professionals working in perinatal healthcare have opportunities to support women in services, to mitigate some of the health risks associated with children's social care involvement and to make appropriate referrals for continued support.[20] Our study aims to explore the experiences of perinatal healthcare providers and reflect on how services and systems might better support women at risk of children's social care involvement (ie, those with 'complex needs').

## METHODS

Survey methods were chosen due to their ability to efficiently collect qualitative data across settings and populations. The anonymity of completing the survey online was also thought to increase honest accounts of professionals' experiences, attitudes and beliefs.[21]

### Sampling strategy

Healthcare professionals were invited to participate via email, newsletters and social media posts on Twitter between October and December 2022. The recruitment process used existing clinical and academic networks, including maternity interest groups and relevant organisations. Participants had to be employed in a healthcare role within perinatal services in England. All responses were voluntary, and consent was sought from each participant. Data were collected via Qualtrics.[22]

### Survey questions

We developed our survey questions based on findings from a systematic scoping review into parental health in the context of children's social care.[10] We asked about

experiences of supporting women with different types of complex health needs that have been associated with children's social care involvement. These included chronic physical conditions, mental health needs, intellectual/developmental disabilities and substance use disorders (see table 1 for definitions). Participants were asked to reflect on their experiences of supporting women with these needs. Most survey questions were free text and had no limit on the length of participants' responses. We asked respondents what they did in their role to support these women and what the barriers to providing care were. We also asked about experiences of safeguarding and referrals to children's social care (see online supplemental material 1 for survey questions).

### Analysis

Descriptive statistics of demographic data were calculated using StatsiQ within the Qualtrics software. Free-text responses were treated as qualitative data and analysed using framework analysis.[23] Authors familiarised themselves with the free-text responses and conducted an initial inductive coding exercise with a subset of data. Phrases were labelled to capture meaning or content relevant to the research question. Initial codes were reviewed and used to develop an analytical framework. Once the framework was refined, all free-text responses from the survey were exported into Excel and data organised within the framework to relevant codes. Codes were conceptually organised to construct themes and capture participant experiences.

### Patient and public involvement

Researchers presented findings to two women with lived experience of children's social care involvement. We asked women to reflect on our themes and descriptions in relation to their own experiences. Minutes from this meeting was considered part of data interpretation and their commentary is included in the results.

### Participants

A total of 70 professionals responded to our online survey, with most participants identifying as women (96%) and white (88%). Half of the respondents worked as midwives (50%) and around half worked in Greater London (46%). See table 2 for the distribution of responders across England. Nearly all had experience of supporting women with mental health conditions (96%),

| Table 1 | Definitions of health need |
|---|---|
| Mental health need | The presentation or diagnosis of symptoms associated with depression, anxiety, psychosis, eating disorders and emotional distress (including post-traumatic stress disorder). |
| Developmental disabilities | An impairment of learning, language or behaviour, inclusive of intellectual (or learning) disabilities, attention deficit and hyperactivity disorder and autism spectrum disorders. |
| Substance use disorders | The use of alcohol, illicit drugs or over the counter or prescription medications in a way that is harmful to the user or those around them. |
| Chronic physical conditions | Conditions that lasted 1 year or more and required ongoing medical attention or limited activities of daily living or both (eg, epilepsy, permanent injuries). |

**Table 2** Participant demographics

| Healthcare professionals | Count (n) | Percentage (%) |
|---|---|---|
| **Gender** | | |
| Female | 67 | 96 |
| Male | 3 | 4 |
| **Age** | | |
| 18–29 | 19 | 27 |
| 30–39 | 18 | 26 |
| 40–49 | 17 | 24 |
| 50+ | 16 | 23 |
| **Ethnicity** | | |
| Asian/Asian British | 3 | 5 |
| Black, Black British, Caribbean or African | 4 | 5 |
| White | 61 | 88 |
| Prefer not to say | 2 | 2 |
| **Job Role** | | |
| Midwife | 35 | 50 |
| Health visitor | 8 | 12 |
| Nurse or support worker | 4 | 6 |
| Perinatal psychologist/psychiatrist | 7 | 10 |
| Other | 16 | 22 |
| **Region of England** | | |
| North East | 1 | 2 |
| Yorkshire and the Humber | 10 | 15 |
| East Midlands | 4 | 6 |
| West Midlands | 11 | 16 |
| South East | 6 | 9 |
| East of England | 2 | 3 |
| South West | 2 | 3 |
| Greater London | 31 | 46 |
| **Years of experience** | | |
| Less than 1 | 11 | 16 |
| 1–5 years | 14 | 20 |
| 5–10 years | 20 | 29 |
| 10–15 years | 10 | 15 |
| 15 years + | 10 | 15 |
| Prefer not to say | 3 | 5 |
| **Direct experience supporting** | | |
| Chronic physical conditions | 37 | 53 |
| Mental health difficulties | 67 | 96 |
| Substance use disorders | 54 | 78 |
| Developmental disabilities | 50 | 72 |
| Child safeguarding concerns | 64 | 92 |

and around three-quarters had supported mothers with substance use disorders (78%), and intellectual/developmental disabilities (72%). Over half had supported a woman with a chronic physical condition (53%) and nearly all had supported a pregnancy that had children's social care involvement (92%) (see table 2).

## RESULTS

From participant data, we constructed five themes: (1) inaccessible healthcare for women with complex needs, (2) the challenges and importance of restoring trust, (3) services focusing on individuals, not families, (4) the necessity and caution around multidisciplinary support and (5) underfunded services inhibiting good practice. Descriptions of each theme and supporting quotations are reported next, alongside commentary from our lived experience group.

### Inaccessible healthcare for women with complex needs

Participants described challenges providing equitable perinatal healthcare for women with complex needs. They gave examples of women with mental health difficulties, disabilities and chronic conditions receiving care in universal clinics which were not suitable for their needs.

> High thresholds for specialist interventions means that the responsibility of ensuring safety of mums and babies is down to us. (ID22, family nurse)

Professionals adjusted care provision as best they could but felt that this was limited without specialist knowledge or facilities. Respondents felt that mothers who faced multiple adversities, including migrant women, often had more complicated pregnancies yet faced additional barriers to care (eg, cultural, language and financial).

> I cared for a woman with a learning disability who was an undocumented migrant…and there was lots of social services involvement around protecting the baby, but none available to support her to keep her child, because of her status. (ID42, midwife)

Midwives stressed the importance of having interpreters at appointments and using resources to aid communication with mothers who had language or communication impairments.

> It's important to use and adapt resources in services, such as birth plans that use images rather than words, or hospital passports to aid communication for those who need it. (ID47, midwife)

Women's access to specialist services was inhibited by lack of local resource, high thresholds and narrow windows of eligibility. For professionals who had experience practising in different areas, the disparity in care quality across England was noted. Participants also described a rise in the number of women using substances throughout pregnancy. They felt that available support was insufficient, especially given that substance use was a prominent feature of safeguarding and risk reports.

> There is a real stigma in both the public and also in maternity services for pregnant women who use

substances. Some services don't even have a specialist substance misuse midwife. (ID07, midwife)

Respondents reflected on how women who used substances were ineligible for support at a mental health 'mother and baby unit', despite mental health needs and substance use likely coexisting. Women could also be excluded from perinatal mental health services if they were under 18, had experienced a pregnancy loss (stillbirth, miscarriage or termination) or had their baby removed at birth.

My clients are mainly under 18, which means that they are unable to access perinatal mental health services and get placed into CAMHS (child and adolescent mental health services) which is generally inappropriate for their needs. (ID39, Family Nurse)

Other services, such as health visiting, halted following newborn removal via children's social care. Professionals commented on this care as fragmented and shortsighted. Our lived experienced group emphasised the importance of being able to access perinatal care, even when those babies were no longer living or had been placed in out-of-home care.

The timeframe they have is so short and once discharged have very little parallel planning for future treatment and support.(ID44, midwife)

Making service provision accessible often required flexibility, such as options for longer or last-minute appointments, communication via SMS, or the offer of home-based support. There was consensus across professional groups that flexible and women-centred perinatal care was the best practice, yet professionals reported feasibility challenges implementing such strategies within National Health Service (NHS) settings.

### The challenges and importance of restoring trust

Respondents stated that building strong relationships with women was key to delivering meaningful care. This included providing emotional support, being an advocate, and treating women with dignity and respect. Our lived experience group felt that professionals should show kindness in their care and be compassionate towards women's circumstances. For psychologists and perinatal mental health therapists, fostering trusted clinical relationships was considered a key feature of treatment and service delivery. Yet this was not always possible for all professionals. Midwives described busy caseloads and staffing issues as barriers to establishing trust with mothers who had complex needs, particularly when there was a lack of continuity of care. The impact of a positive relationship on engagement with services and mother/baby outcomes was noted.

Relationship building is the key. A woman will return for advice if she feels understood and listened to. I am in a much better position to support and advice if

a woman trusts me and understands what I say to her. (ID11, health visitor)

For women at risk of children's social care involvement, professionals felt that there was an ingrained distrust of services. Respondents reflected on having to restore women's confidence in public services' ability to support them. Considerations of how to build back trust included validating women's experiences, providing a non-judgmental space to listen, and understanding aspects of appointments which might be difficult (eg, discussing safeguarding concerns and potential outcomes).

Remaining open and non-judgemental to women. To remain a person that they can trust and to help anchor them into health care services. Support them with social care processes whilst giving them realistic expectations of what is likely to happen. (ID62, midwife)

Our lived experience group agreed that honest and clear communication helped with any escalation of safeguarding concerns.

Participants stated the importance of being able to provide support for families and not just appearing to be policing them. They stated that if a family do not feel they are being heard or supported, they will be more likely to withhold information about health behaviours that trigger concern, such as substance use. Yet the support professionals provided was constrained by service rigidity and safeguarding policies. Some midwives provided examples of strained services being risk averse and inhibiting meaningful risk management with families experiencing difficulties. Referrals to internal safeguarding teams, domestic abuse pathways or children's social care, at times lead to further distrust of services for families already under stress.

A woman disclosed domestic violence and although I explained that I had to make a referral I was worried that I would lose her trust. This is what happened—she was contacted by social services in a very unhelpful way and was furious. She declined any further care from me. (ID33, midwife)

Responding to safeguarding concerns sensitively and proportionately relied on trusted clinical relationships between professionals, as well as with families.

Often, we are not getting feedback on safeguarding referrals, the information all seems to flow one way. So, we are not always feeling that our voices, as midwives, are being heard when social services get involved. (ID39, midwife)

### Services focusing on individuals, not families

Risk to the unborn baby was a main consideration for determining maternal care plans and pathways, often limiting more holistic, family-focused work.

"Services may or may not take a rounded view of the mother and her child's need." (ID22, psychologist),

as "the unborn baby or child's safety can often take priority over efforts to safeguard the family as a whole." (ID54, midwife)

A participant (psychologist) stated that services should support both mother and baby, offering services that target women's needs as a means of enabling caregiving capacity, and therefore supporting newborn safety. Midwives and health visitors commented on the importance of looking to existing networks of support in a woman's life to support risk management. Involving partners was described as integral to crisis planning, yet support became more complicated when needs were interrelated (eg, domestic abuse or cohabiting drug use). Participants described how household risks were often outside the remit of perinatal services, despite raising safeguarding concerns for the baby.

> Often safeguarding concerns are related to fathers' needs. A woman's partner having severe mental health problems, not engaging with services, being abusive in the house and the woman being subject to coercive control. (ID62, clinical nurse specialist)

Our lived experience group reflected on how services should be mindful of any potential abuse in households and work to support the whole family in a way that does not further the risk to the mother. One of the health visitors commented on the dearth of services available for birth fathers. While, in principle, some perinatal services supported both parents, all were targeted at mothers and most had little capacity to support the wider family unit.

For some women, their own childhood experiences and relationships with their parents played a negative role in their transition to motherhood. Two of the psychology respondents described the perinatal period as a sensitive time for managing complex trauma and felt services should be equipped to manage this.

> There are not enough resources to help women with severe trauma through the several years needed to make it safe/viable for them to be good and creative mothers to their children when they had insufficient support themselves when a child. (ID22, psychologist)

These participants recommended that any 'whole family' support considers both families of origin and families of creation. Our lived experience group also commented on services acknowledging children in families who are no longer alive or have been placed in out-of-home care.

### The necessity and caution around multidisciplinary support

Participants reported liaising and working alongside other professionals to support women with complex needs yet described challenges to shared care plans.

> Social services use a different system to us, so information sharing is not always done timely. It's even difficult to find schedules for all to be available at the same time. (ID03, occupational therapist)

However, participants cautioned that routine data sharing across sectors could inhibit mothers' engagement with services.

> Pressure from other services, such as children's social care, make it difficult for women to engage meaningfully with mental health services, making the support superficial. There is fear that services feedback to social care and influence child protection proceedings. (ID52, staff nurse)

Participants identified high thresholds for specialist services as a barrier to multidisciplinary working, with referrals being rejected or mothers being added to long waiting lists. Midwives referenced specialist mental health services as especially hard to access, with many women left without support.

> I have had a lot of women struggle to access mental health services due to long waiting lists, sometimes longer than their pregnancies. I have also had many women rejected because their mental health condition was not severe enough. (ID48, midwife)

Liaising with housing, benefits and similar services was viewed as integral to supporting women's health. Respondents mentioned that local and third-sector offerings should be a part of signposting and referral processes. This was corroborated by the lived experience group, who appreciated professionals having local knowledge of organisations that might help specific needs. Yet survey responses illustrated how staff often felt unprepared when supporting women in situations, or with conditions, that they had not received training on. Participants acknowledged that services being overstretched meant less time to dedicate to 'over and above' care, such as building a knowledge base of local provision, learning about more unusual conditions or engaging with optional training. This left staff feeling less able to fulfil their role; for example, some health visitors and midwives described being unable to provide breastfeeding advise to women on certain medications.

> Not everyone understands the full extent of conditions. Staff at services dealing with chronic physical health conditions won't know the impact it has on pregnancy, and maternity staff don't know the full extent of how pregnancy can impact on chronic physical health conditions. (ID36, occupational therapist)

Our lived experience group spoke about the importance of understanding the 'whole picture' and not to simply focus on needs which fit within the remit of a specific service.

### Underfunded services inhibiting good practice

Most notably, midwives reported insufficient staff levels and an overwhelming number of referrals and appointments. All professionals reported concern around the lack of resourcing for staff and services, alongside an increasing demand of complex pregnancies.

Midwives are all incredibly busy due the national shortage of midwives. Women working with children's social care require additional midwifery input which many feel they do not have capacity to facilitate. (ID22, midwife)

Midwives also felt that training on specific health needs was insufficient, especially for temporary bank staff brought in due to staffing shortages. One participant reflected on training courses focusing on how to refer to specialist midwives, but not how to support these women in regular antenatal clinics. They reported feeling worried about not knowing how to answer mothers' questions or how to best tailor care.

Having too many babies and women on our caseloads with specialist needs, without being trained to be able to support them. This is a growing area of occurrence and NHS services need to increase their resources and staffing to meet these growing demands. (ID66, midwife)

Staff across professions and services reported high rates of burn out, with unsustainable staffing levels and expectations. Many reflected on how being unable to adequately support mothers had left them feeling frustrated and disheartened. For some respondents, there were comments on how vicarious trauma had impacted on their own well-being and ability to fulfil their duties of care.

It is a very emotionally draining job and often can be very sad. There is a lack of adequate resourcing and capacity issues within the services themselves. (ID08, midwife)

Most participants referenced good leadership, management and clinical supervision as key components to fostering a healthy workforce.

Effective supervision is one of the key ways in which we can help to keep babies safe and support their mothers. (ID52, psychologist)

Responses spoke to wider fatigue experienced by public service staff navigating public spending cuts and austerity measures.

## DISCUSSION

Our findings describe experiences of perinatal healthcare professionals who support women with complex needs at risk of children's social care involvement. In keeping with wider literature, our results illustrate shortcomings in care provision for women experiencing multiple disadvantages.[9] Given the recent independent review into children's social care,[24] and the push for targeted, early help for vulnerable families, this study offers a timely insight into the role of perinatal services.

Healthcare professionals shared ideas for adaptations and service-level changes that would improve perinatal equity for these mothers, including resourcing professionals to deliver family-focused, multidisciplinary care which is compassionate and proportionate to need. This is consistent with what mothers involved with children's social care have reported elsewhere, a need for comprehensive support which addresses multiple disadvantages and enables safe parenting.[20 25 26] Yet despite agreement between mothers and professionals of what 'good' perinatal healthcare looks like, our findings highlight various barriers to implementing this in practice.

Our results indicate that despite the will of individual practitioners, systemic issues within the NHS can result in women at risk of children's social care receiving inadequate perinatal healthcare.[19] Maternity services in England continue to be an area of concern for the Care Quality Commission, with services and staff under huge pressure to maintain safe clinical standards.[27] In order to provide meaningful support to vulnerable families in the antenatal period, professionals and services must be properly resourced.[28] For example, developing trust with mothers was important for delivering effective care in this context[29]; however, professionals described challenges doing so in services that were overstretched, understaffed and managing increasingly complex caseloads.[24] Our results draw attention to the dedication of perinatal healthcare staff who were often doing the best they could, with the resources they had, to meet the needs of vulnerable families. Yet we know that increasing numbers of frontline staff are leaving due to exhaustion and burnout.[30] In keeping with the NHS Long Term Plan, our findings highlight the need to drive recruitment into professional training pathways, such as midwifery and psychology,[31] as well as retain staff in public healthcare, through adequate resourcing, good leadership and opportunities for professional development.[32]

We found that rigid service structures can leave women who experience multiple disadvantages in the perinatal period falling between the gaps of provision.[9 19] Without targeted intervention during pregnancy, women who are vulnerable to children's social care involvement are not given a fair opportunity to promptly address their difficulties.[9] As illustrated in our findings, this can be frustrating for professionals who often feel unable to provide effective care, or hold family-level risk, within perinatal services. As a result, women can experience fragmented care which is not always responsive to level of need, for example, women being 'left behind' by perinatal services following mother–baby separation, despite the known consequences on maternal health.[10 33 34]

There are local examples of good practice, including continuity of care models,[35] prebirth support led by specialist midwives[36] and interventions to support women in perinatal services following newborn separation.[37] However, such initiatives are not underpinned by national policy or guidelines, leading to inconsistent provision for women across the country.[38] Previously published antenatal guidelines from 2010 (last reviewed in 2018) offer advice for supporting women with

**Table 3** Suggested updates to NICE guidelines by research theme

| Research theme | Current NICE guidelines | Suggested update |
|---|---|---|
| Inaccessible healthcare for women with complex needs | "At the booking appointment, give the woman a telephone number to enable her to contact a healthcare professional outside of normal working hours, for example the telephone number of the hospital triage contact, the labour ward or the birth centre." | 1. Service leads to provide oversight on tailored care provision for women experiencing multiple disadvantages, including flexible delivery and continuity of care<br>2. To deliver specialist perinatal healthcare in circumstances where mothers 'fall between the gaps' of services (eg, migrant women, mothers <18 years old, those who use substances or have lost an infant to death or to care). |
| The challenges and importance of restoring trust | "Respect the woman's right to confidentiality and sensitively discuss her fears in a non-judgemental manner."<br>"Addressing women's fears about the involvement of children's services and potential removal of their child, by providing information tailored to their needs" | 1. To understand the impact of past and ongoing trauma on families' interactions with services and remain empathetic when addressing non-engagement<br>2. To be transparent about service capabilities and local safeguarding protocols; cultivating systems where vulnerable families feel heard and supported, rather than policed and judged. |
| Services focusing on individuals, not families | "In order to facilitate discussion of sensitive issues, provide each woman with a one-to-one consultation, without her partner, a family member or a legal guardian present, on at least one occasion." | 1. Consider how services and third-sector organisations might support 'whole-family' need, including domestic abuse, and birth fathers' mental health or substance use<br>2. To work with mothers, and families, to build risk management plans which draw on their strengths and existing networks of support. |
| The necessity and caution around multidisciplinary support | "Consider initiating a multi-agency needs assessment, including safeguarding issues, so that the woman has a coordinated care plan"<br>"Tell the woman why and when information about her pregnancy may need to be shared with other agencies." | 1. To develop guidelines around data-sharing which are sensitive and proportionate to risk (ie, between perinatal mental health services and children's social care)<br>2. To encourage ongoing communication between agencies after referrals, with feedback and shared decision-making across children's social care and perinatal services. |
| Underfunded services inhibiting good practice | "Healthcare professionals should be given training to ensure they are knowledgeable about safeguarding responsibilities for both the young woman and her unborn baby." | 1. Resource perinatal healthcare services with safe staffing levels, sustainable workloads and sufficient equipment to promote equitable care delivery<br>2. Increase training, further education and supervision capacity for healthcare professionals and encourage career development. |

'complex social factors'.[39] Yet these guidelines do not address multiple and intersectional disadvantages, or the impact of maternal trauma relevant to service provision.[9] More recent evidence has indicated that women at risk of children's social care involvement require targeted and specialist perinatal care.[20 40] Our results support the need for updated national guidelines. Some recommendations are outlined in table 3.

### Strengths and limitations of this study

We collected survey data across eight regions of England and a range of professional groups. Free-text responses allowed professionals to share, in confidence, detailed responses and case examples of service provision and constraints. However, these data are also limited in scope and diversity, given that most respondents were white, worked as midwives and were based in Greater London. It should be acknowledged that survey methods can be biased towards those who 'self-select' to participate and have a given motive for doing so (eg, a strong view that they wish to share). More detailed understanding

of experiences across professional groups and local authority areas is needed to inform service improvement in this area.

### CONCLUSION

Women who are at risk of children's social care involvement will likely experience perinatal healthcare inequities. Our findings suggest that current perinatal healthcare provision for this population is inadequate and national guidelines should be updated.

**Author affiliations**
[1]Department of Epidemiology and Public Health, University College London Institute of Epidemiology and Health Care, London, UK
[2]Homerton Healthcare NHS Foundation Trust, London, UK
[3]Division of Methodologies, Florence Nightingale Faculty of Nursing Midwifery & Palliative Care, King's College London, London, UK
[4]Population, Policy and Practice, University College London Great Ormond Street Institute of Child Health, London, UK
[5]Population Health Research Institute, St George's University of London, London, UK

[6]Thomas Coram Research Institute, University College London Social Research Institute, London, UK

**Acknowledgements** We would like to thank the healthcare professionals who took the time to complete our online survey. We would also like to acknowledge the support of our project advisory group of women who offered support in the development of this study and in interpreting findings.

**Contributors** CG, JW, CP, RMB and RL conceived the idea for the study. CG conducted the data collection and analysis. TB-M and BLT assisted with data interpretation and manuscript editing. CG drafted the manuscript, and all authors approved the final draft. CG accepts full responsibility for the finished work and the conduct of the study as guarantor, has access to the data, and controlled the decision to publish.

**Funding** This work was supported by the Economic and Social Research Council UBEL Doctoral Training Programme (ES/P000592/1) and a UCL Culture Engagement Grant. CP, JW and RL are (in part) supported by the National Institute for Health and Care Research (NIHR) Children and Families Policy Research Unit (PR-PRU-1217–21301).

**Disclaimer** The views expressed are those of the author(s) and not necessarily those of the NIHR or the Department of Health and Social Care.

**Competing interests** None declared.

**Patient and public involvement** Patients and/or the public were involved in the design, or conduct, or reporting, or dissemination plans of this research. Refer to the Methods section for further details.

**Patient consent for publication** Consent obtained directly from patient(s).

**Ethics approval** This study received ethical approval from University College London's Research Ethics Committee (ref: 21605/001). Participants gave informed consent to participate in the study before taking part.

**Provenance and peer review** Not commissioned; externally peer reviewed.

**Data availability statement** Data are available upon reasonable request.

**ORCID iDs**
Claire Grant http://orcid.org/0000-0002-1545-6428
Billie Lever Taylor http://orcid.org/0000-0002-6865-3425
Claire Powell http://orcid.org/0000-0002-6581-0165
Ruth Marion Blackburn http://orcid.org/0000-0002-3491-7381
Jenny Woodman http://orcid.org/0000-0002-9403-4177

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
