## [Reviewer comments · BMJ Open]

ARTICLE DETAILS

TITLE (PROVISIONAL)	Perinatal healthcare for women at risk of children's social care involvement: a qualitative survey of professionals in England
AUTHORS	Grant, Claire; Bicknell-Morel, Tamsin; Lever Taylor, Billie; Powell, Claire; Blackburn, Ruth; Lacey, Rebecca; Woodman, Jenny

VERSION 1 – REVIEW

REVIEWER	Nyikavaranda, Patrick Brighton and Sussex Medical School, Primary Care and Public Health
REVIEW RETURNED	19-Dec-2023

GENERAL COMMENTS	A good and thought-out research. I wondered though if Table 3 Suggested updates to NICE guidelines by research theme" may have been better suited as a supplementary file. Just an opinion but something you can ignore especially if you are not limited to word count. Also, please re-check proofing, for example, line 333, is it advise or advice? I assume you meant advice in the context. Please check grammar throughout. Loved the inclusion of a lived experience commentary complementing the analysis as part of the results section. Very well done and brings a new insight to the research.
---

REVIEWER	Gowda, Mamatha Jawaharlal Institute of Postgraduate Medical Education and Research
REVIEW RETURNED	01-Jan-2024

GENERAL COMMENTS	First of all would like to extend my appreciation to all the researchers and authors for conducting this study on an important social subject. However, the draft reads long though very well written. It would help retain the interest of readers if the manuscript can be made little more concise. 1. Background Section paragraphs needs to be improved, some sentences like "Line 88- We know that histories of trauma....may be changed to it is known that or studies have shown that...as this is based on a reference and not on personal opinion-rephrasing such sentences recommended. 2. The Results section is long...while retaining the excerpts of the interviews/questionnaire, other content in result such as opinions of researchers can be expressed in a more concise manner. Reduction of Results section by 10-15% recommended.
--

VERSION 1 – AUTHOR RESPONSE

Comments from reviewer #1:

A good and thought-out research. I wondered though if Table 3 Suggested updates to NICE guidelines by research theme" may have been better suited as a supplementary file. Just an opinion but something you can ignore especially if you are not limited to word count. Also, please re-check proofing, for example, line 333, is it advise or advice? I assume you meant advice in the context. Please check grammar throughout. Loved the inclusion of a lived experience commentary complementing the analysis as part of the results section. Very well done and brings a new insight to the research.

Response from authors:

Many thanks for your time reading this manuscript and for the supportive comments provided. The authors have reviewed the document to check for grammatical errors and have edited accordingly.

Given that there is allowance in the word count, we have decided to keep the table of NICE recommendations in the main body of the text. The justification for this is that we hope findings of this survey will have some direct clinical implications, and the table will likely be of interest to readers who work in a clinical capacity. We hope that they can be integrated into the main messages conveyed by the paper, rather than an addendum.

Reviewer #2:

First of all, would like to extend my appreciation to all the researchers and authors for conducting this study on an important social subject. However, the draft reads long though very well written. It would help retain the interest of readers if the manuscript can be made little more concise.

Response from authors:

Many thanks for your positive comments and the time you have taken to review this paper. We have considered your comments and respond to each below.

Comment:

Background Section paragraphs needs to be improved, some sentences like "Line 88- We know that histories of trauma....may be changed to it is known that or studies have shown that...as this is based on a reference and not on personal opinion-rephrasing such sentences recommended.

Response from authors:

The authors appreciate that the use of 'we' suggests an opinion, rather than an evidence-based statement. As such, we have edited these statements to 'it is known'.

Comment:

2. The Results section is long...while retaining the excerpts of the interviews/questionnaire, other content in result such as opinions of researchers can be expressed in a more concise manner. Reduction of Results section by 10-15% recommended.

Response from authors:

Many thanks for your comment. We agree that the results section is long. As mentioned, much of this is due to the qualitative nature of the survey and involvement of quotations from participants. We have edited to remove some researcher commentary to shorten the length slightly and hope this is sufficient (see updated manuscript).